# Sorbent Based on Polyvinyl Butyral and Potassium Polytitanate for Purifying Wastewater from Heavy Metal Ions

**Anna Ermolenko** [1], **Maria Vikulova** [1], **Alexey Shevelev** [1], **Elena Mastalygina** [2,3], **Peter Ogbuna Offor** [4], **Yuri Konyukhov** [5], **Anton Razinov** [2], **Alexander Gorokhovsky** [1] **and Igor Burmistrov** [1,2,5,*]

[1] Department of Chemistry and Technology of Materials, Yuri Gagarin State Technical University of Saratov, 77 Polytecnicheskaya Street, 410054 Saratov, Russia; molish01@mail.ru (A.E.); vikulovama@yandex.ru (M.V.); titans5@rambler.ru (A.S.); algo54@mail.ru (A.G.)

[2] Educational and Scientific Center "Trade", Plekhanov Russian University of Economics, 36 Stremyanny Lane, 117997 Moscow, Russia; elena.mastalygina@gmail.com (E.M.); razinov.ae@rea.ru (A.R.)

[3] Emanuel Institute of Biochemical Physics, Russian Academy of Sciences, 4 Kosygina str., 119334 Moscow, Russia

[4] Metallurgical and Materials Engineering Department, University of Nigeria, Nsukka 410001, Nigeria; peter.offor@unn.edu.ng

[5] Department of Functional Nanosystems and High-Temperature Materials, National University of Science & Technology "MISIS", 4 Leninsky pr., 119049 Moscow, Russia; ykonukhov@misis.ru

**\*** Correspondence: glas100@yandex.ru; Tel.: +7-91-7201-8703

**Abstract:** Currently, the rapid development of industry leads to an increase in negative anthropogenic impacts on the environment, including water ecosystems. This circumstance entails toughening environmental standards and, in particular, requirements for the content of pollutants in wastewater. As a result, developing technical and cost-effective ways for wastewater purification becomes relevant. This study is devoted to the development of a novel composite sorbent, based on polyvinyl butyral and potassium polytitanate, designed to purify water from heavy metal ions. The co-deposition of a mixture based on a polymer solution and a filler suspension was used to obtain a composite material. In this work, the influence of the deposition conditions on the structure and properties of the resulting composites was studied, as well as the optimal ratio of components, including solvent, precipitant, polymer binder, and filler, were established. In the course of this study on the sorption properties of the developed composite materials using various sorption models, the sorption capacity of the obtained material, the sorption mechanism, and the limiting stage of the sorption process were determined. The developed sorbent can be suitably used in the wastewater treatment systems of galvanic industries, enterprises producing chemical current sources, and in other areas.

**Keywords:** polymer composite; sorbent; wastewater; polyvinyl butyral; potassium polytitanate; heavy metals

---

## 1. Introduction

The increase in environmental pollution is one of the main problems of modern society. The number of various pollutants, including heavy metal ions [1], toxicants and dyes [2,3], bacteria, and viruses, entering the wastewater system is rapidly increasing due to the rapid development of industry and population growth. In this regard, the problem of water purification becomes more relevant, which is reflected in the number of studies devoted to this topic.

Heavy metal ions are not destroyed under environmental factors, and, on the contrary, they tend to accumulate in various living organisms. Getting involved in complex food webs, heavy metals adversely affect plants and animals and end up in the human diet. Owing to this, the pollution of wastewater by heavy metal ions is a problem of increasing importance for environmental, evolutionary, food, and environmental reasons [4].

There is a wide variety of ways employed in treating wastewater for heavy metal ions, including chemical precipitation [5], ion exchange [6], reverse osmosis [7], electrolysis [8], membrane filtration [9], and adsorption [10–13]. The listed methods have both advantages and disadvantages. Among the technologies implemented in practice, ion-exchange sorption is one of the most effective and cost-effective methods of removing heavy metals from aqueous media [14–16]. This water treatment method provides a high degree of purification (close to 100%) for highly polluted water. The possibility of sorbent regeneration can be realized with a relatively simple hardware design. The most common sorption materials for heavy metals are synthetic zeolite, natural kaolinite, chitin, chitosan [17–21], carbon materials [22–24], and agricultural waste, such as rice bran and orange peels [25]. According to recent studies, synthetic nanostructured carbon sorbents [18,26] have a very high sorption capacity (up to 97 mg of $Pb^{2+}$ per gram of sorbent) [27], which makes them one of the most interesting objects of research in this field. Nevertheless, there are several disadvantages of carbon sorbents that limit their widespread and routine use. The disadvantages include high cost, as well as limited volumes and the complexity of waste product disposal [28]. Natural sorbents based on layered aluminosilicates (kaolinite, montmorillonite, zeolite, and their analogues) are quite cheap and widespread. However, their sorption properties depend significantly on many external factors, such as deposits and impurities, pH level, temperature, the presence of either or both electrolytes and surfactants, etc. [13,29]. Hence, the need for the complex chemical and thermal modification of the feedstock before practical use.

Among synthetic sorption materials with a layered structure, potassium titanates show a high sorption capacity, and good sintering into glass–ceramic composites, whereby they can be effectively processed into safe and valuable ceramic products [30]. The sorption characteristics of some crystalline modifications of potassium titanate were studied in the works [31,32]. For instance, the sorption capacity of crystalline potassium hexatitanate ($K_2Ti_6O_{13}$) for $Cd^{2+}$ ions can be up to 0.80 mmol/g [31], and the sorption capacity of crystalline potassium tetratitanate ($K_2Ti_4O_9$) for $Cu^{2+}$ ions can be up to 1.94 mmol/g [32]. Amorphous potassium titanates demonstrated effectiveness in the sorption of organic dyes [33] and $Ni^{2+}$ ions [34]. In our previous work, the sorption of $Pb^{2+}$ and $Sr^{2+}$ ions by the amorphous potassium titanate was studied. It was shown that the sorption capacity for $Pb^{2+}$ ions was 714 mg/g and for $Sr^{2+}$ ions it was 345 mg/g [35].

To avoid the problem of separating the filtrate from the purified solution for powder catalysts, a number of techniques can be used, for example, granulation, coating on substrates, creating membranes, and the formation of polymer composites. In order to develop composite sorbents, carbon nanotubes, lignin, graphene oxide, chitosan, concrete, and others, are used. When creating composites with sorption properties, a number of difficulties arise, including the dispersion of filler in a polymer matrix and the filler pre-treatment, which ensures the availability of particles of the filler (sorbent) for ions from the solution.

Previously, polyvinyl butyral (PVB) was studied as a polymer matrix for obtaining composites with sorption properties [30]. Nevertheless, the sorption composite was used in the form of a thin-film coating, which is convenient for photocatalytic systems studied by the authors but is ineffective for the absorption of soluble ions from the water stream.

In this regard, the aims of this study were to create a composite based on potassium polytitanate and polyvinyl butyral with open porosity for the effective sorption of heavy metal ions from a solution and to analyze its sorption capacity and mechanism. These aims were achieved by creating a porous highly filled composite by a coagulation precipitation technique previously used for polypropylene-based composites [36]. Isopropyl alcohol was used as a solvent for PVB, and distilled water was used as a precipitant. The dependence of the structure of the composites and properties on the deposition

conditions were studied. The sorption mechanism was determined for the composites with an optimal structure (large open porosity, high degree of filling).

## 2. Materials and Methods

### 2.1. Materials

Potassium polytitanates (PPT) were synthesized in a hydroxide-salt melt, according to the procedure described in [37]. As the raw components, titanium dioxide $TiO_2$ (99% anatase, Aldrich, Darmstadt, Germany, CAS 13463-67-7), potassium hydroxide KOH (98% purity, Vekton, St. Petersburg, Russia), and potassium nitrate $KNO_3$ (98% purity, Reahim, Moscow, Russia), in a mass ratio of 30:30:40, were used. The components were blended with distilled water in a ratio of water/titanium dioxide = 2:1, then exposed to thermal treatment in a muffle furnace for 3 h at a temperature of 500 °C.

Polyvinyl butyral (hereinafter PVB), of the film-cast type (PP grade, GOST 9439-85) and supplied by "VitaKhim" LLC (Moscow, Russia), was used as a polymer binder for the composites. To prepare the PVB solution, the dehydrated isopropyl alcohol (hereafter IPA) (GOST 9805-84), supplied by "Zavod sinteticheskogo spirta" CJSC (Orsk, Russia), was used.

Standardized test solutions for the study of sorption capacity were made of lead nitrate (high grade, GOST 4236-77, Russia) and manufactured by NPF "Nevsky Khimik" (St. Petersburg, Russia).

### 2.2. Preparing a Highly Filled Porous Composite Based on Potassium Polytitanates

PVB was dissolved in isopropyl alcohol at an elevated temperature of 90 °C by a mechanical stirrer operating at a speed of 150 rpm. Simultaneously, a suspension of PPT in isopropyl alcohol was prepared by stirring with a mechanical stirrer (200 rpm) in a ratio of 1, 3, or 5 g of PPT to 50 mL of alcohol (in order to ensure the different filling degree of composites). After dissolving PVB, the solution was combined with the suspension of PPT (to obtain the concentrations shown in Table 1) and mixed for 15 min by a mechanical stirrer (150 rpm). Then the resulting mass was poured (thin stream) into an excess of precipitant (water) where rapid coagulation occurred. The volume ratio of the PVB solution (mixed with PPT powder) to the precipitant (water) was 1:2, 1:4, or 1:6. After formation, the composite was removed from the water–alcohol solution by decantation and dried at 20 °C for 24 h.

**Table 1.** The compositions of porous composite materials based on potassium polytitanate.

| N° | Sample Notation | PVB * Content in Composite, wt.% | PPT ** Content in Composite, wt.% | Ratio of Isopropyl Alcohol/Water |
|---|---|---|---|---|
| 1 | 1/6-70-30 | 70 | 30 | 1/6 |
| 2 | 1/4-70-30 | 70 | 30 | 1/4 |
| 3 | 1/2-70-30 | 70 | 30 | 1/2 |
| 4 | 1/4-50-50 | 50 | 50 | 1/4 |
| 5 | 1/4-90-10 | 90 | 10 | 1/4 |

* Polyvinyl butyral. ** Potassium polytitanates.

By a process of rapid coagulation of PVD, when alcohol is mixed with water, a sponge-like composite was formed. The type of porosity of the resulting composite depended on the volume ratio of the suspension to the precipitant, as well as the content of PPT in the initial suspension. A wide range of PPT concentrations and several different ratios of isopropyl alcohol/water were studied to provide the maximum concentration of PPT (because only PPT sorbs heavy metal ions, but PVB is just a passive matrix holding titanate) and to keep some strength of the sponge-like composite (so that the PPT-based sorbent was not crumbled during operation, cutting, or pouring into different containers).

The surface morphology of the composites was studied by a US-made Aspex Explorer scanning electron microscope (SEM) with an energy dispersion attachment.

### 2.3. Sorption Capacity Assessment and Kinetic Analysis

To determine the dynamic sorption characteristics of the composites, the installation shown in Figure 1 was used.

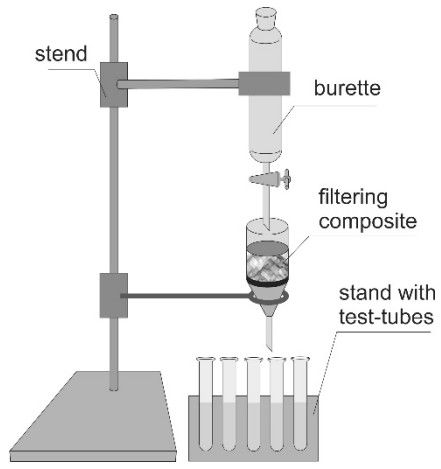 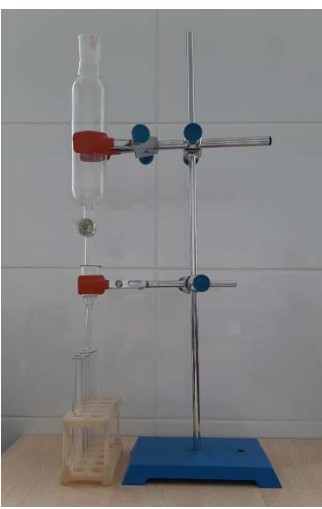

**Figure 1.** Installation for the study of the process of the dynamic sorption of lead ions.

The solution of lead nitrate (75 g/L), with a constant speed (6.5 mL/min), was fed into the funnel, in which there was the porous polymer composite. The filtrate was collected in 10 mL portions. Furthermore, the obtained materials were studied using X-ray fluorescence analysis, which was carried out using a Spectroscan-MAX-G spectrometer (LLC "NPO" SPECTRON, Russia, St. Petersburg) with a scanning crystal-diffraction channel. Gauge curves were previously found to determine the concentration.

The values obtained by changes in the concentration of lead ions over time were evaluated using the following models: Boyd model, Lagergren pseudo-first order kinetic model, Ho and McKay's pseudo-second order kinetic model, and Weber–Morris intraparticle diffusion model.

The Boyd model allows the evaluation of the role of diffusion in the sorption process. For this model, development Equation (1) was used [38]:

$$F = \frac{Q_t}{Q_e} = 1 - \left(\frac{6}{\pi^2}\right)\exp(-Bt), \tag{1}$$

where $F$ = degree of process completion, $Q_t$ = quantities of ions at current time (mg/g), $Q_e$ = quantities of ions in a state of sorption quasi-equilibrium (mg/g), which are adsorbed at time $t$ and have a state of sorption quasi-equilibrium, respectively.

The Lagergren Equation (2) describes the pseudo-first order kinetic model for sorption processes [39]:

$$\mathrm{Lg}(Q_e - Q_t) = \mathrm{lg}Q_e - \frac{k_1}{2303}t, \tag{2}$$

where $k_1$ = rate constant (min$^{-1}$).

To consider the kinetics of the sorption process by a pseudo-second order model, a Ho and McKay Equation (3) was used [40]:

$$\frac{t}{Q_t} = \frac{1}{k_2 Q_e^2} + \frac{t}{Q_e}, \tag{3}$$

where $k_2$ = rate constant (g·mg$^{-1}$·min$^{-1}$).

A Weber–Morris intraparticle diffusion model was expressed by the following Equation (4) [41]:

$$q_t = K_{dif} t^{0.5} + C_i, \tag{4}$$

where $K_{dif}$ = intraparticle diffusion rate constant (mg·g$^{-1}$·min$^{-0.5}$), and $C_i$ = parameter characterizing the thickness of the boundary layer (mg/g).

## 3. Results and Discussion

To determine the effects of the volume ratio of the suspension/precipitant and PPT content in the initial suspension on the composite structure, a range of samples were used (Table 1). The choice was based on the desire to create a composite with a maximum concentration of PPT, but at the same time, so that it did not crumble during operation, cutting, or pouring into different containers.

The appearance of the cross-section of the studied composite is shown in Figure 2. The cross-section was obtained manually by slicing blades. There was a significant decrease in the porosity of the composite materials (Figure 2a–c), that could be caused by a decrease in the amount of precipitant. When the isopropyl alcohol/water ratio was 1:2 (Figure 2c), most of the internal pores were insulated (almost no open porosity remained) and, in turn, the outer surface of the composite became smooth. The composite sample 1/4-50-50 (Figure 2d) had a low porosity compared to the sample 1/2-70-30. In this case, this is due to the high filler content (mass ratio filler/matrix = 1:1). In the last studied composite (Figure 2e), the amount of potassium polytitanate introduced into the PVB was the smallest, which led to the immersion and isolation of PPT particles in the composite matrix, and as a result of which their activity, the surface was not available for physicochemical processes.

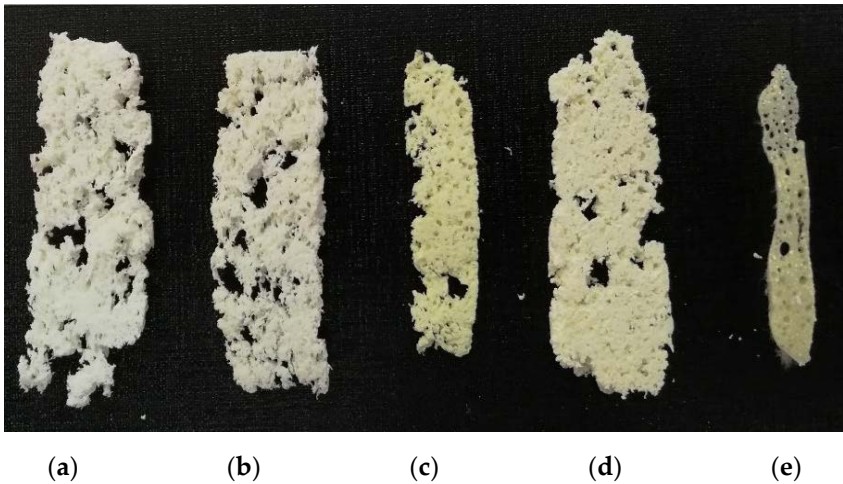

|  (a) | (b) | (c) | (d) | (e) |

**Figure 2.** The slices of the obtained porous composite materials: (**a**) 1/6-70-30; (**b**) 1/4-70-30; (**c**) 1/2-70-30; (**d**) 1/4-50-50; (**e**) 1/4-90-10.

Figure 3 presents microphotographs of the composite samples obtained by scanning electron microscopy (SEM). The filling degree and the conditions for obtaining them significantly affected the porosity of the materials. The composite containing 10 wt.% of PPT (sample 1/4-90-10) had a dense structure with a predominantly closed porosity. When 50 wt.% of the filler was introduced into the polymer matrix, the composite was characterized by a loose structure that could be easily destroyed by the slightest mechanical stress.

With a change in the volume ratio of the PPT suspension to the precipitant, the pattern of micropores changed with a change in macropores (Figure 3). For the first two samples (1/6-70-30 and 1/4-70-30), the pore size was 20–30 μm and 5–10 μm, respectively. The third sample (1/2-70-30) was characterized by a significant decrease in the number of micropores and the formation of closed pores (size did not exceed 5 μm). For sample 1/6-70-30, the presence of PPT particles localized in pores and a

lack of moistening with a binder was observed (Figure 3a). Moreover, an increase in the PTT content in the initial suspension (sample 1/4-50-50) led to the predominant localization of PPT in the form of individual particles that did not interact with the binder. The observed distribution of PPT particles could provide the most effective sorption, but, obviously, could lead to the chipping of the sorption filler from the composite.

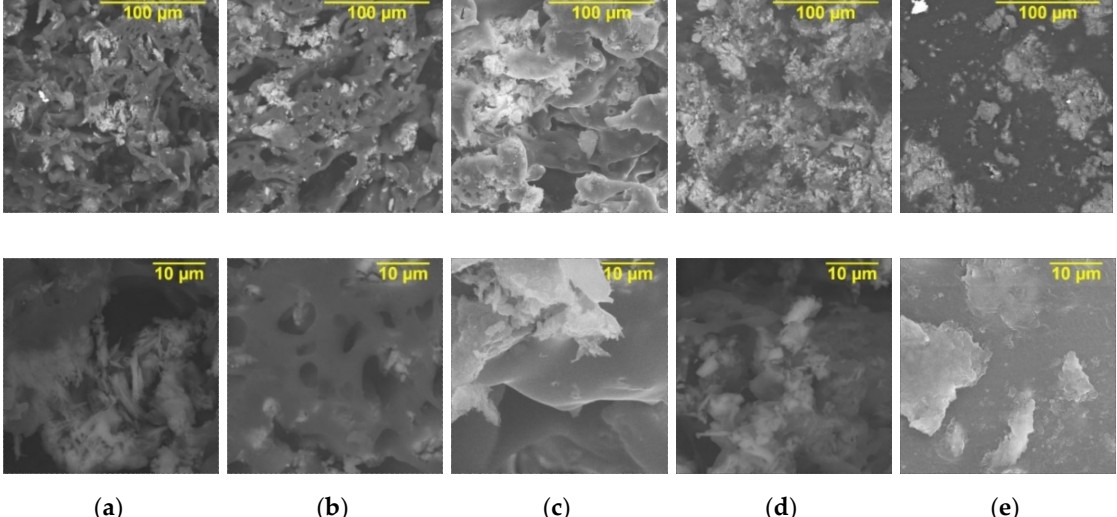

|      (a)      |      (b)      |      (c)      |      (d)      |      (e)      |
| :---: | :---: | :---: | :---: | :---: |

**Figure 3.** The microphotographs of the composites obtained by SEM: (**a**) sample 1/6-70-30 based on 70 wt.% of PVB and 30 wt.% of PPT, IPA/water = 1:5; (**b**) sample 1/4-70-30 based on 70 wt.% of PVB and 30 wt.% of PPT, IPA/water = 1:4; (**c**) sample 1/2-70-30 based on 70 wt.% of PVB and 30 wt.% PPT, IPA/water = 1:2; (**d**) sample 1/4-50-50 based on 50 wt.% of PVB and 50 wt.% of PPT, IPA/water = 1:4; (**e**) sample 1/4-90-10 based on 90 wt.% of PVB and 10 wt.% of PPT, IPA/water = 1/4. Abbreviations: SEM—scanning electron microscopey; PVB—polyvinyl butyral; PPT—potassium polytitanates; IPA—isopropyl alcohol.

The reduction in the amount PPT in the initial suspension (sample 1/4-90-10) led to a noticeable aggregation (most likely due to the increased viscosity of the suspension). While the filler particles were completely wetted by the matrix, which could increase the mechanical properties of the materials, this could limit access to filtered solutions and adversely affect sorption characteristics.

The sample 1/4-90-10 obviously cannot be an effective sorption material, primarily because of the low content of the active component in the composite. In addition, PPT particles were almost completely immersed in the polymer matrix, which blocked them from interacting with heavy metal ions. The composite 1/4-50-50, on the contrary, contained the largest amount of filler. However, its further use was limited by a low degree of fixation of PPT particles by the polymer binder, which created the probability of the dispersed powder getting into the purified solution, and, in turn, the necessity of the additional cleaning of the solution. The samples with a ratio of PPT/PVB = 30:70 were considered optimal, since the concentration of an active component was sufficient for effective sorption interaction with an optimal ratio with a binder, which ensured the effective fixation of particles of potassium polytitanate, while maintaining the maximum available surface area. However, the high viscosity of the solvent in the case of a dilution of 1/4 and 1/2 had a significant effect on the micropore size, reducing them by more than three to four times, which was one of the determining factors for evaluating the sorption ability.

According to the study of the composite structure, it was assumed that the sample 1/6-70-30, containing 70 wt.% of PVB and 30 wt.% of PPT (IPA/water = 1:6), had the best sorption ability. Therefore, the sorption of lead ions by this material was studied under dynamic conditions.

It is necessary to give an isotherm describing the dependence of the concentration on time and show the sorption capacity. The effect of the contact time in the range of 1.5–32.0 min on the lead ion sorption by composite materials is shown in Figure 4.

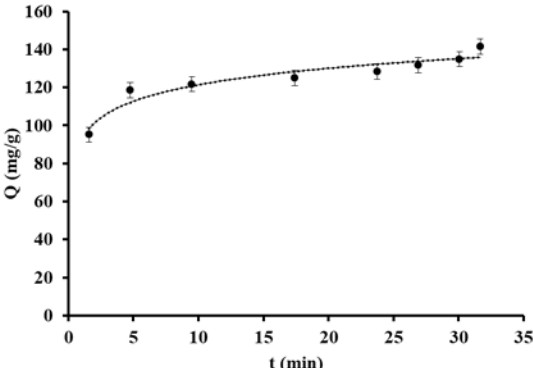

**Figure 4.** Effect of contact time on the lead ion sorption by the porous polymer composite of 1/6-70-30.

According to the kinetics curve, the sorption of lead ions by the studied polymer composite was a quick process. After 5 min of interaction with the polymer composite, most of the lead ions were removed from the solution. A highly concentrated solution (0.23 mol/L) and a high filtration rate (6.5 mL/min) were used to study the sorption kinetics. Under these conditions, the residual concentration was 0.09 mol/L. The sorption capacity of the composite began to increase slowly when the time increased from 10 to 30 min. The porous structure of the composite and available active sites of potassium polytitanate provided the high sorption rate at the beginning of the process. When the surface of the sorbent was covered and blocked, the sorption capacity value stopped changing. A little more than 30 min was needed to attain the equilibrium with a sorption capacity of 141.5 mg/g.

The role of diffusion during sorption was evaluated using the Boyd model. Figure 5a shows the time-varying behavior of the parameter $-\ln(1-F)$, which estimates the contribution of external diffusion to the overall process rate during the sorption of lead ions.

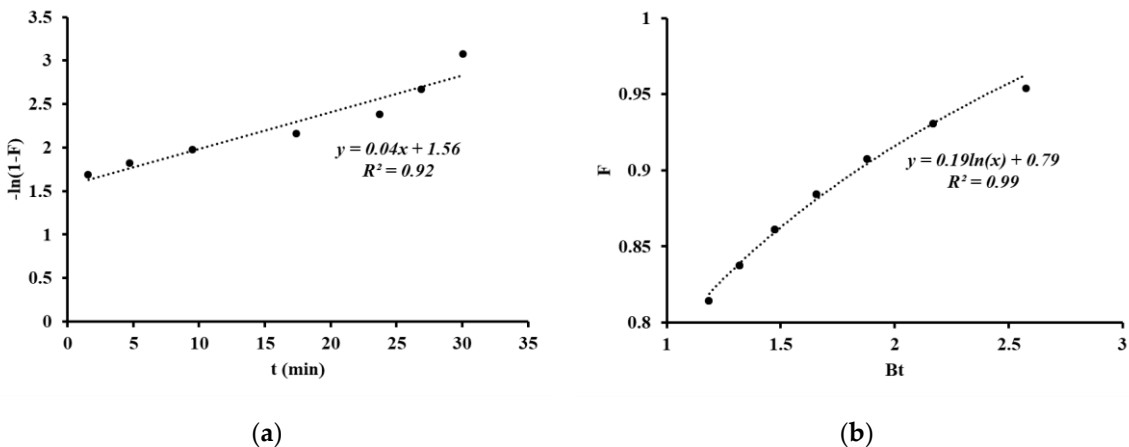

(a)                                                              (b)

**Figure 5.** The dependence of $-\ln(1-F)$ on time (*t*) (**a**) and sorption degree (*F*) on *Bt* parameter (**b**) for the sorption of lead ions by the porous polymer composite of 1/6-70-30.

According to the calculated coefficient of determination ($R^2 \sim 0.9$), during the sorption of lead ions by the porous composite, the external processes occurred, but they were not limiting. It should be noted that the plot of the dependence of the $-\ln(1-F)$ parameter on time did not pass through the origin of coordinates, which indicates the mechanism of the mixed diffusion of sorption.

In Figure 5b, the contribution of an internal diffusion to the sorption rate was estimated by the dependence of *F*–*Bt*. The high coefficient of determination ($R^2$) of the existing dependence between

the experimental and calculated data of the logarithmic trend line indicates the high role of internal diffusion during the sorption of lead ions by the porous polymer composite.

The Weber–Morris model takes into account the diffusion at the phase boundary and inside the sorbent particles. Figure 6 shows the kinetics of sorption defined according to this model, using Equation (4).

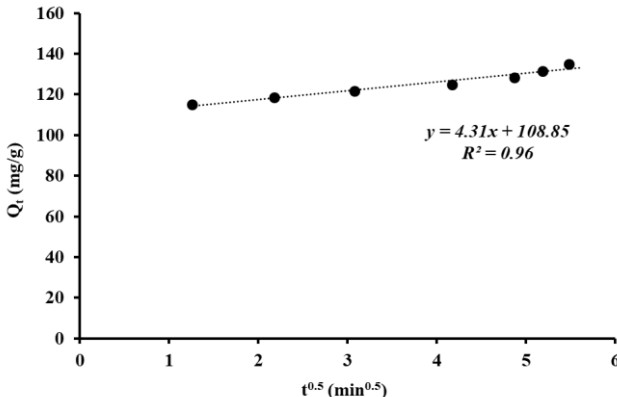

**Figure 6.** The kinetics of sorption by the porous polymer composite of 1/6-70-30 in the coordinates of the Weber–Morris model.

The existing dependence of $Q_t$ on $t^{0.5}$ fits well on the straight line, which is described by the equation $y = 4.31x + 108.85$. It indicates that the rate of the sorption process was determined by the diffusion inside the sorbent particles. Thus, composite 1/6-70-30 had enough macroporosity and microporosity to provide an effective sorption, limited by the processes inside the aggregates of PPT particles, not by the rate of access to them. However, the linear plots did not pass through the origin. Such behavior of line could be due to the difference in the rate of mass transfer in the various sorption stages and some degree of boundary layer limits. Based on the obtained equation of linear dependence, diffusion parameters were calculated. The parameter of the boundary layer thickness ($C_i$) was equal to 108.85 mg/g, which confirmed the rapid sorption in a short period. The value of the intraparticle diffusion rate constant ($K_{dif} = 4.31$ mg·g$^{-1}$·min$^{-0.5}$) indicates a rapid diffusion process.

The chemical kinetics models of pseudo-first and pseudo-second orders describe the contribution of chemical reactions to the sorption rate. The correspondence of the sorption processes to the pseudo-first order kinetic model was estimated using the Lagergren Equation (2). Figure 7 shows the kinetics of sorption, according to the pseudo-first (Figure 7a) and the pseudo-second (Figure 7b) order models.

Since the coefficient of determination was lower than 0.95, the contribution of diffusion at a stage not preceding sorption was not significant. The pseudo-second order model was constructed using the Ho and McKay Equation (3). The high value of the determination coefficient indicates the correspondence of this process to the pseudo-second order model and can be described by the equation $y = 0.0074x + 0.0057$, by which the kinetic parameters were determined. The implication is that the stage of chemical interaction of lead ions and PPT is a limiting one during the sorption ($k_2 = 0.006$ g·mg$^{-1}$·min$^{-1}$, $Q_e = 137.0$ mg/g). According to this model, the interaction of the sorbent and sorbate obeys the law of the mass action. Therefore, the rate of a chemical reaction is directly proportional to the concentration of two substances reacting in a ratio of 1:1.

Table 2 shows the comparison of the sorption capacity of the developed polymer composite based on polyvinyl butyral and potassium polytitanate (sample 1/6-70-30) with the same parameter as other polymer composites with respect to lead ions. According to the data, the sorption capacity of the developed sorbent based on polyvinyl butyral and potassium polytitanate is third only to those of composites based on poly (3,4-ethylenedioxythiophene)/lignin and poly (N-vinylcarbazole)/graphene oxide.

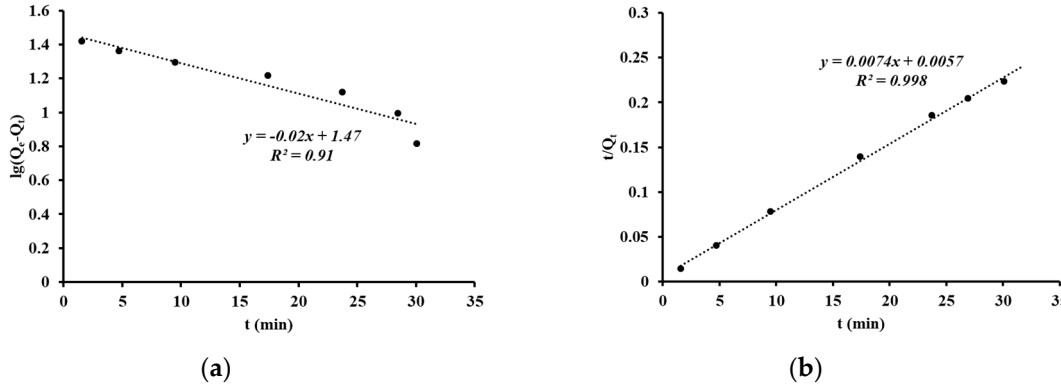

**Figure 7.** The kinetics of sorption by the porous polymer composite of 1/6-70-30 in the coordinates of the pseudo-first (**a**) and (**b**) the pseudo-second order models.

In the case of the composite obtained by Checkol F. et al. [42], at a high efficiency, even after 60 cycles (47% of the sorption capacity was retained), obtaining a composite film was rather labor-intensive by galvanostatic deposition. The degradation of the composite developed by Karnib M. et al. [43] was not studied, while the necessity of activating the filler surface by heat treatment and drying in a vacuum was indicated, which requires additional equipment. Vacuum drying was also required when manufacturing the composites in the other research [44,45]. The rest of the composites are characterized by a low sorption capacity to lead ions in comparison with the material developed in this work. Moreover, these composites are characterized by complex manufacturing technologies with preliminary functionalization of the filler surface [44–47], in situ polymerization [44], or using irradiation and a nitrogen atmosphere [48]. For a number of composites [44,46,47,49], the efficiency of removing lead ions after three to four cycles of the process did not decrease significantly, and for the others this parameter was not studied. In addition, the high cost of the composite materials due to the cost of polymers or fillers should be noted; in some cases, both components cost several thousand or tens of thousands of roubles.

**Table 2.** The sorption capacity of different polymer matrix composites with respect to lead ions.

| Polymer Composite | Sorption Capacity (mg/g) | Reference |
|---|---|---|
| Polyvinyl butyral/potassium polytitanate (1/6-70-30 sample) | 137.0 | - |
| Potassium polytitanate | 714.3 | [35] |
| Poly (3,4-ethylenedioxythiophene)/lignin | 452.8 | [42] |
| Polypyrrole/multi-wall carbon nanotube | 26.3 | [44] |
| Poly (ethylene imine)/silica gels | 82.6 | [49] |
| Ion-imprinted polymer Pb(II)/SBA-15 | 42.6 | [46] |
| Poly-methacrylic acid grafted chitosan-bentonite nanocomposite | 111.0 | [48] |
| Cellulose/TiO$_2$ | 120.5 | [47] |
| Carboxymethyl-β-cyclodextrin/Fe$_3$O$_4$ | 64.5 | [45] |
| Poly (N-vinylcarbazole)/graphene oxide | 982.9 | [43] |

The most interesting activity is the comparison of the results obtained for the composite polyvinyl butyral/potassium polytitanate with the results for pure potassium polytitanate, which was studied in our previous work [33]. The sample 1/6-70-30 had the PVB/PPT ratio of 70/30. For this sample, if the PPT sorption capacity is fully realized (assuming that the PVB sorption capacity is negligible), the sorption capacity of the composite will be 30% of the PPT capacity (214 mg/g). However, according to the experimental data, the sample 1/6-70-30 was characterized by the sorption capacity being equal to 137.0 mg/g. Obviously, the polymer matrix hindered access to the PPT sorbent (filler), thus reducing the sorption capacity. Nevertheless, the sorption capacity of the developed polymer composite was about 70% of the sorption capacity of the pure PPT.

## 4. Conclusions

The polyvinyl butyral/potassium polytitanate composite sorbent for lead ions was investigated. The influence of the composite production conditions on the sorption properties of the developed composite materials was studied.

The sorption mechanism was determined using various kinetic models. The sorption process of lead ions by the polyvinyl butyral/potassium polytitanate composite was confirmed by a pseudo-second order kinetic model with a better correlation compared with the pseudo-first. It meant that both the concentrations of sorbate and sorbent were the factors determining the process rate. Additionally, this kinetic model showed that the potential rate-controlling stage of the sorption was a chemisorption with cation exchange and the participation of potassium cations in either or both the PPT interlayer space reaction and the complexation reaction on the sorbent surface. The calculated value of $Q_{e\ (calculated)}$ = 137.0 mg/g deviated slightly from the experimental one, $Q_{e\ (experimental)}$ = 141.5 mg/g. The conformity of the experimental data to diffusion models was evidence that the external mass transfer was of little importance. Hence, the intraparticle diffusion was significant in the sorption rate determination for the system of PVB/PPT–Pb(II), especially at the later stages of this process. A Boyd kinetic plot confirmed that the internal mass transfer was the slowest step involved in the sorption process.

Due to the influence of the polymer matrix, which impedes the access of lead ions from the solution to the sorbent (filler), the sorption capacity of polytitanate was just partially realized. When chemical particles interact freely and the optimal parameters needed for the preparation of the composites are in place, the sorption capacity of polytitanate can be up to 70% of the value of pure filler. Based on the results obtained, the polyvinyl butyral/potassium polytitanate composite sorbent can be used for the treatment systems of wastewater containing lead ions. For example, for treatment of wastewater from ore-dressing factories and mines, metallurgical plants, and many chemical industries, such as the for the production of batteries, glass, paints, insecticides, and gasoline, etc.

**Author Contributions:** Conceptualization, A.G. and I.B.; methodology, A.E.; software, Y.K. and A.R.; validation, I.B., A.S. and A.G.; formal analysis, A.G.; investigation, A.E., M.V. and A.S.; resources, A.G. and I.B.; data curation, A.S.; writing—original draft preparation, E.M. and P.O.O.; writing—review and editing, E.M. and P.O.O.; visualization, M.V. and A.R.; supervision, I.B.; project administration, Y.K. and A.S. All authors have read and agreed to the published version of the manuscript.

**Funding:** This research received no external funding.

**Conflicts of Interest:** The authors declare no conflict of interest.

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
