# Peer review of "Sorbent Based on Polyvinyl Butyral and Potassium Polytitanate for Purifying Wastewater from Heavy Metal Ions"

_processes, doi:10.3390/pr8060690_

Round 1

Reviewer 1 Report

The authors have presented a paper in which they compare the elaborations, characteristics and performance of different materials in terms of its sorption capacity. The topic is interesting and related to the journal. Nonetheless, the paper has some problems that have to be addressed before considering its publication. Following, I have included a list of problems that I have found:

In general terms in the paper, there are a lot of small paragraphs. Authors have to consider to merge some of those short paragraphs.

I suggest removing the acronyms from the abstract

At the end of the introduction, the authors have to include the aim of the paper in a short paragraph.

After the aim of the paper, the authors have to detail the structure of the rest of the paper.

The authors have to check the format of the paper. Some paragraphs have a different format than others.

Obtained equations related to the kinetics of sorption must be included in the paper.

The authors have to justify in the material and methods how they decide and define the different compositions used.

In Figure 2, authors have to explain the differences found in all the combinations and how they are explained.

Axe of Figure 5 b) is not complete.

I do not understand why in the material and methods authors detail 5 different composts, and at the beginning of section 3 the images of the 5 composts are shown, but then, authors only present the kinetics of just one of the composts.

In additions, I do not understand why in all the graphics authors present data of 7 measured points (from 1.5 tp 30 min) and in Figure 7 (b) the authors include 20 measured points (from 1.5 to up to 30 min).

In Table 2, authors have to include the Sorption capacity of all tested materials.

A discussion of the results must be included in the paper. Authors have compared the Sorption capacity of one of the proposed composite with the data found in the literature. However, more comparisons must be made. Other parameters (not only the sorption capacity) have to be compared, such as degradation, complexity to obtain the composite, price, etc.

Author Response

Dear Reviewer,

the authors are grateful for the careful consideration, interesting and important comments! All comments are carefully corrected, corrections are highlighted in color.

Reviewer comments: 

The authors have presented a paper in which they compare the elaborations, characteristics and performance of different materials in terms of its sorption capacity. The topic is interesting and related to the journal. Nonetheless, the paper has some problems that have to be addressed before considering its publication. Following, I have included a list of problems that I have found:

In general terms in the paper, there are a lot of small paragraphs. Authors have to consider to merge some of those short paragraphs.

- Thank you for pointing out this. Small paragraphs in the introduction were merged.

I suggest removing the acronyms from the abstract

- The acronyms were removed from the abstract.

At the end of the introduction, the authors have to include the aim of the paper in a short paragraph.

- The aim of the paper was added in a short paragraph at the end of the introduction.

After the aim of the paper, the authors have to detail the structure of the rest of the paper.

- The short details of the rest of the paper have been added: Achieving these aims were carried out by creating a porous highly filled composite by coagulation precipitation technique previously used for polypropylene-based composites. Isopropyl alcohol was used as a solvent for PVB, and distilled water was used as a precipitant. The dependence of the composites structure and properties on the deposition conditions was studied. For the composites with an optimal structure (large open porosity, high degree of filling) the sorption mechanism was determined. Line 91-96

The authors have to check the format of the paper. Some paragraphs have a different format than others.

- All the formats were checked and corrected.

Obtained equations related to the kinetics of sorption must be included in the paper.

- The approximation equation from the graph in Fig. 6 was used to find out the coefficient of the Weber-Morris model; the equation from the graph in Fig. 7b was used to find out the coefficient of the pseudo-second order models. Additionally, the approximation equations were added for all the graphs of sorption kinetics. We agree that they can be of interest to readers and improve the presentation of data. Clarifying comments are added to the text: line   254-255, 260-261 and 273-274.

The authors have to justify in the material and methods how they decide and define the different compositions used.

- The choice is based on the necessity to create a composite with a maximum concentration of PPT, because only PPT sorbs heavy metal ions, but PVB is just a passive matrix holding titanate (in order to PPT was not crumbled during operation, cutting, pouring into different containers). Additional explanations are included in the manuscript: line 125-130

In Figure 2, authors have to explain the differences found in all the combinations and how they are explained.

- The explanation was added to the section “Results and discussion”, line 171-175.

Axe of Figure 5 b) is not complete

- For convenient perception of the graphical dependence, only fragments of the axis are given.

I do not understand why in the material and methods authors detail 5 different composts, and at the beginning of section 3 the images of the 5 composts are shown, but then, authors only present the kinetics of just one of the composts.

- The description of the structure of different composites was added under Figure 3. For effective sorption of ions from a solution, the composite must have a maximum degree of filling, open porosity, and good filler availability for adsorbed ions. Using SEM, we determined which material best meets these requirements (it is the sample 1/6-70-30) and it is this material that has been investigated to find out the sorption mechanism, more detailed explanations are added to the text: line 204-216.

In additions, I do not understand why in all the graphics authors present data of 7 measured points (from 1.5 tp 30 min) and in Figure 7 (b) the authors include 20 measured points (from 1.5 to up to 30 min).

- This measurement was performed several times due to doubts about the interpretation of the data. Therefore, there were more points in comparison with the rest of the figures. For the convenience of readers and a clear comparison of figures, the extra point was removed from the figure 7b.

In Table 2, authors have to include the Sorption capacity of all tested materials.

- The study of the sorption capacity of all tested composites has no technical significance, since the SEM results show that their structure is not suitable for sorption applications. Some explanations are presented in the text: line 204-2016.

A discussion of the results must be included in the paper. Authors have compared the Sorption capacity of one of the proposed composite with the data found in the literature. However, more comparisons must be made. Other parameters (not only the sorption capacity) have to be compared, such as degradation, complexity to obtain the composite, price, etc.

- The comparison of the proposed composite with the data found in the literature was performed. The comparison with the data found in the literature in various parameters has been improved, please, see lines 287-300 and in Table 2.

Reviewer 2 Report

The aim of the article is clear to prepare a composite for efficient sorption of Pb based on PPT and

PVB and to analyze its sorption capacity and mechanism. The preparation method of composite base is clear and reproducible. The analysis of it is correct. But as the original base, i.e. only potassium titanate sorbent has much more higher sorption capacity, a detailed explain would be needed for the last sentence of the article,  "the PVB/PPT composite sorbent can be used for the treatment systems of waste water containing lead ions."

 90          „ineffective for absoption..” is it absorption or adsorption….?

202         we support the claim that:”most of the ions were removed from
               the solution.

243         „Since the coefficient of determination was lower than 0.9” - which
               figure the statement  applies to? Fig.7(a) R2=0.9125

Author Response

Dear Reviewer,

the authors are grateful for the careful consideration, interesting and important comments! All comments are carefully corrected, corrections are highlighted in color.

Reviewer comments: 

The aim of the article is clear to prepare a composite for efficient sorption of Pb based on PPT and PVB and to analyze its sorption capacity and mechanism. The preparation method of composite base is clear and reproducible. The analysis of it is correct. But as the original base, i.e. only potassium titanate sorbent has much more higher sorption capacity, a detailed explain would be needed for the last sentence of the article,  "the PVB/PPT composite sorbent can be used for the treatment systems of waste water containing lead ions."

- Thank you for pointing out this because it is really can be explained in more detail. The text has been added to line 333-335: For example, for treatment of wastewater from ore-dressing factories and mines, metallurgical plants, many chemical industries, such as the production of batteries, glass, paints, insecticides, gasoline, etc.

90          „ineffective for absoption..” is it absorption or adsorption….?

- In this case, we mean “absorption”.

202         we support the claim that:”most of the ions were removed from the solution.

- Thank you for your deep understanding of the work! A highly concentrated solution (0,23 mol / l) and a high filtration rate (6.5 ml / min) were used to study the sorption kinetics. Under these conditions, the residual concentration was 0.09 mol / l. Of course, reducing the speed or increasing the amount of sorbent will increase the degree of purification significantly. However, the residual content of lead ions is not a fundamental characteristic of the material, but, on the contrary, can be considered as a matter of technology development. Relevant additions are included in the text: line 228-230 and line 137

243         „Since the coefficient of determination was lower than 0.9” - which figure the statement applies to? Fig.7(a) R2=0.9125

- This is our misprint. The mistake was corrected.

Round 2

Reviewer 1 Report

Authors have adequately addressed all my previous comments